# Hyperuricemia—Especially “Metabolic Hyperuricemia”—Is Independently Associated with a Higher Risk of Steatotic Liver Disease

**DOI:** 10.3390/metabo15060356

**Published:** 2025-05-28

**Authors:** Janis Timsans, Jenni Kauppi, Vappu Rantalaiho, Anne Kerola, Kia Hakkarainen, Mika Paldanius, Hannu Kautiainen, Markku Kauppi

**Affiliations:** 1Department of Rheumatology, Päijät-Häme Central Hospital, Wellbeing Services County of Päijät-Häme, 15850 Lahti, Finlandmarkku.kauppi@paijatha.fi (M.K.); 2Faculty of Medicine and Health Technology, Tampere University, 33100 Tampere, Finland; vappu.rantalaiho@tuni.fi; 3Unit of Physiatry and Rehabilitation Medicine, Päijät-Häme Central Hospital, Wellbeing Services County of Päijät-Häme, 15850 Lahti, Finland; jenni.kauppi@paijatha.fi; 4Centre for Rheumatic Diseases, Tampere University Hospital, 33521 Tampere, Finland; 5Department of Medicine, Kanta-Häme Central Hospital, 13530 Hämeenlinna, Finland; 6Institute for Molecular Medicine Finland, Helsinki Institute of Life Science, University of Helsinki, 00014 Helsinki, Finland; 7Department of Nephrology, Päijät-Häme Central Hospital, Wellbeing Services County of Päijät-Häme, 15850 Lahti, Finland; kia.hakkarainen@paijatha.fi; 8Oulu University of Applied Sciences, 90570 Oulu, Finland; mika.paldanius@oamk.fi; 9Folkhälsan Research Center, 00250 Helsinki, Finland; hannu.kautiainen@medcare.fi; 10Primary Health Care Unit, Kuopio University Hospital, 70029 Kuopio, Finland; 11Clinicum, Faculty of Medicine, University of Helsinki, 00014 Helsinki, Finland

**Keywords:** hyperuricemia, serum uric acid, serum urate, metabolic hyperuricemia, renal hyperuricemia, steatotic liver disease, fatty liver, FLI (fatty liver index), NAFLD (nonalcoholic fatty liver disease), MASLD (metabolic dysfunction associated steatotic liver disease)

## Abstract

**Background**: Hyperuricemia and steatotic liver disease are both associated with various comorbidities and mortality. This study was carried out to study the association between hyperuricemia and steatotic liver disease and to assess the impact of the etiology of hyperuricemia on the development of steatotic liver disease. **Methods**: Data from a population-based study of 2635 individuals aged 52–76 years was used. Hyperuricemia was defined as a serum urate (SU) of >410 μmol/L (≈6.9 mg/dL; 75th percentile) and steatotic liver disease as a fatty liver index (FLI) of ≥80 (75th percentile). We defined hyperuricemia as renal if estimated glomerular filtration rate (eGFR) was ≤67 mL/min/1.73 m^2^ (25th percentile) and as metabolic if eGFR was >67 mL/min/1.73 m^2^. **Results**: FLI correlated with SU in women [r = 0.47 (95% CI: 0.43 to 0.51)] and men [r = 0.37 (95% CI: 0.32 to 0.42)]. Compared to those with SU ≤ 410 μmol/L and FLI < 80, the hazard ratio (HR) for all-cause mortality was 1.76 (95% CI: 1.39 to 2.23) in hyperuricemic individuals with FLI ≥ 80, 1.16 (95% CI: 0.95 to 1.40) in hyperuricemic individuals with FLI < 80, and 1.34 (95% CI: 1.06 to 1.70) in persons with SU ≤ 410 μmol/L and FLI ≥ 80. Individuals with metabolic hyperuricemia had a statistically significantly higher FLI than individuals with renal hyperuricemia: mean (SD) = 73.4 (12.2) and 69.6 (22.5), respectively, *p* = 0.015 after adjusting for sex and diabetes. **Conclusions**: FLI correlates positively with SU, and it is higher in persons with metabolic hyperuricemia. Both steatotic liver disease and hyperuricemia increase mortality.

## 1. Introduction

Gout is a treatable chronic painful inflammatory arthritis caused by the accumulation of monosodium urate crystals in joints and periarticular tissue [1]. Hyperuricemia is a prerequisite for the development of gout but not all hyperuricemic individuals develop gout, and most people with elevated serum urate (SU) levels remain asymptomatic [2]. Although asymptomatic hyperuricemia does not lead to gout attacks, the condition is nevertheless harmful—it is independently associated with several disorders (e.g., hypertension, chronic kidney disease, coronary artery disease, and diabetes) [3] and with mortality [4,5,6,7,8,9,10,11,12,13].

Uric acid is synthesized primarily in the liver from hypoxanthine in reactions catalyzed by xanthine oxidoreductase (XOR) and excreted from the body mainly by the kidneys (and partially also by the intestine) [14]. If there is a disturbance in either of those mechanisms (excessive production or ineffective excretion), SU rises. According to a previously published paper more than 90% of gout patients underexcrete and less than 10% overproduce urate [15]. Around 10% of the urate filtered by the glomeruli is excreted in the urine, while the rest is reabsorbed in the proximal tubule. Urate transport in the proximal tubule occurs in both directions, involving both reabsorption and secretion. Increased reabsorption and decreased secretion play a role in etiology of hyperuricemia. Urate reabsorption is primarily mediated by the transporters URAT1, GLUT9, and OAT4, while secretion is mainly facilitated by OAT1, ABCG2, NPT1, and NPT4 [16].

The differences in the risk of comorbidities and mortality related to the different etiologies of hyperuricemia have not been studied previously. A distinction between hyperuricemia due to underexcretion and hyperuricemia due to overproduction has been made neither in epidemiological studies nor in trials on the effect of urate lowering therapy (ULT). In a previous study, however, we have studied if the hyperuricemia-related mortality risk differs by degree of renal function impairment. Although hyperuricemia and kidney impairment have an additive effect on mortality [17], our findings indicate that hyperuricemia in individuals with normal renal function (metabolic hyperuricemia) is associated with a significantly higher all-cause mortality risk than hyperuricemia related to impaired kidney function (renal hyperuricemia) [18]. This association is even more pronounced in regard to the cardiovascular mortality risk effect related to hyperuricemia [19]. These findings indicate that metabolic hyperuricemia may pose a bigger threat to health than renal hyperuricemia. The suggestion that hyperuricemia may be more detrimental in the absence of renal impairment than when accompanied by reduced kidney function is also supported by findings from the Uric Acid Right for Heart Health (URRAH) study, where a high serum uric acid to creatinine ratio was associated with increased cardiovascular risk [20].

Steatotic liver disease is also an important and mortality increasing health risk [21] and is associated with several comorbidities [22]. The prevalence of steatotic liver disease has been rising for some decades and it appears that this trend will continue in the years to come [23]. Some reports have found that steatotic liver disease is associated with elevated SU [24,25,26].

In the present study, we aimed to examine the association between hyperuricemia and the risk of steatotic liver disease, and whether this association differs between individuals with renal versus metabolic hyperuricemia. Additionally, we investigated mortality rates among individuals with and without hyperuricemia and steatotic liver disease. Our central hypothesis was that serum urate (SU) is independently associated with hepatic steatosis, and that this association is stronger in individuals with metabolic hyperuricemia compared to those with renal hyperuricemia.

The preliminary findings of this study were presented at ACR Convergence 2023 [27].

## 2. Materials and Methods

### 2.1. Study Population

We used data from the GOAL (Good Aging in Lahti region) study—a prospective, population-based study of elderly people in the catchment area of the Päijät-Häme central hospital (located in Lahti, Finland). The recruitment period for this study began on 1 February 2002, and ended on 31 December 2002. Individuals from three age cohorts (52–76 years) were invited—persons born in 1926–1930, 1936–1940 and 1946–1950). The baseline visits took place in 2002, and follow-up visits were conducted in 2005, 2008 and 2012. Mortality data were available up until the end of 2018. A total of 4272 subjects were invited and 2815 (66%) responded to the invitation. In the final analysis, only individuals with no missing relevant laboratory values were included, resulting in a total of 2635 participants. The flowchart detailing the selection of study subjects is shown in Figure 1.

Data collected from each study participant included serum urate levels and various blood parameters (creatinine, cystatin C, blood glucose, total cholesterol, LDL-C, HDL-C, triglycerides, C-reactive protein [CRP], high-sensitivity CRP, and 25-hydroxyvitamin D). Additional information was gathered on socioeconomic status, psychosocial background, education, income, lifestyle habits (such as smoking, alcohol use, and physical activity), and previously diagnosed medical conditions (including hypertension, diabetes, coronary heart disease, stroke, and cancer). Medication use and hospitalization records from the 12 months preceding the baseline visit were also documented. Blood pressure was measured three times at baseline, with the average recorded. Height, weight, and waist circumference (measured midway between the lowest rib and the iliac crest) were assessed, and body mass index (BMI) was calculated. Urinary parameters (e.g., urinary urate and urinary creatinine) were not available.

For the purposes of this study the data were accessed on 11 April 2023. The authors of this study did not have access to information that could identify individual participants.

We calculated the fatty liver index (FLI) from the triglyceride level, BMI, gamma-glutamyl transferase (GGT) activity and waist circumference using the algorithm published by Bedogni et al. [28]:

FLI = (e^0.953 × loge (triglycerides) + 0.139 × BMI + 0.718 × loge (ggt) + 0.053 × waist circumference − 15.745^)/(1 + e^0.953 × loge (triglycerides) + 0.139 × BMI + 0.718 × loge (ggt) + 0.053 × waist circumference − 15.745^) × 100. The units of measurement used in the equation are: mg/dL for triglycerides, kg/m^2^ for BMI, U/L for GGT, and cm for waist circumference. The value of the FLI ranges from 0 to 100.

FLI is an accurate predictor of hepatic steatosis in the general population, and it is one of the most relevant tools to detect uncomplicated steatosis in the stage where fibrosis has not yet developed [29]. It has been validated in multiple different cohorts [30,31,32,33]. The higher the FLI, the higher is the likelihood of hepatic steatosis. FLI value of 30 and 60 are used as cutoff values: FLI < 30 rules out hepatic steatosis with a sensitivity of 87% and FLI ≥ 60 rules in hepatic steatosis with specificity of 86%. As many as 96% of persons with FLI ≥ 80 have steatotic liver [28]. FLI has also been shown to be associated with arterial stiffness [34] and to serve as a strong predictor of cardiovascular disease risk over 10-year follow-up periods [35,36].

In our study we present persons with SU > 410 μmol/L (≈6.9 mg/dL; 75th percentile) as clearly hyperuricemic and persons with FLI ≥ 80 (75th percentile) as having steatotic liver. We also conducted a subgroup analysis among clearly hyperuricemic individuals based on the etiology of hyperuricemia (renal vs. metabolic). In line with previous studies [18,19], we defined renal hyperuricemia as serum urate elevation in individuals with an estimated glomerular filtration rate (eGFR) ≤ 67 mL/min/1.73 m^2^ (25th percentile), and metabolic hyperuricemia as elevated serum urate in those with an eGFR > 67 mL/min/1.73 m^2^. The glomerular filtration rate was calculated using the CKD-EPI creatinine-cystatin C equation, which is a very accurate marker for estimating renal function in elderly individuals [37].

Mortality data were provided by Statistics Finland. The causes of death were classified according to the International Statistical Classification of Diseases and Related Health Problems, 10th Revision (ICD-10). Alcohol-related causes of death included all diseases caused by alcohol (ICD-10 categories F10, G31.2, G40.51, G62.1, G72.1, I42.6, K29.2, K70, K85.2, K86.0, O35.4, P04.3 and Q86.0). The follow-up of each subject started at the time of the first study visit (2/2002 to 8/2002) and ended on 31 December 2018.

### 2.2. Statistical Methods

The characteristics of the participants were expressed as means with standard deviations (SD), or as counts with percentages. The relationship between FLI and SU was evaluated using a generalized linear models (e.g., analysis of covariance and logistic models) with appropriate distribution and link function. Sidak multiple comparison procedure was used to correct significance levels for post hoc testing (α 0.05), when appropriate. Cox proportional hazards regression was used to estimate the crude and adjusted hazard ratios (HR) and their 95% confidence intervals (CIs); models included age, sex, education, smoking status, alcohol consumption, body mass index, hypertension, dyslipidemia and diabetes as covariates. The proportional hazard assumption was tested graphically and by using a statistical test based on the distribution of Schoenfeld residuals. Crude and adjusted estimates of mortality per 1000 person–years were calculated using Poisson regression models. The ratio of observed to expected number of deaths, the standardized mortality ratio (SMR) for all-cause deaths, was calculated using subject–years methods with 95% confidence intervals. The expected number of deaths was calculated on the basis of sex-, age- and calendar-period-specific mortality rates in the Finnish population (Official Statistics of Finland). Correlations were estimated by Pearson’s correlation coefficient method. The Stata 18.0, StataCorp LP (College Station, TX, USA) statistical package was used for the analyses.

### 2.3. Ethics

The study followed the guidelines of the Declaration of Helsinki. The cohort study was approved in 2002 by the Ethics Committee of Päijät-Häme Central Hospital, which is located in the city of Lahti (ID number of approval: PHSP 2/2002/Q11 § 87). All participants gave their written informed consent prior to data collection. The use of the data gathered in the cohort study combined with registry data were approved by HUS Regional Committee on Medical Research Ethics in 2019 (ID number of approval: HUS/1748/20219 § 124).

### 2.4. Patient and Public Involvement

Patients and/or public were not involved in the design, conduct, reporting or dissemination plans of this research.

## 3. Results

A total of 2815 persons responded to the invitation. The SU level and the variables for calculating the FLI were available from 2635 individuals (1386 women, 1249 men (47%)), representing 94% of all participants. The mean age of the participants was 64 years (range 52–76 years). The total person–years was 38,313, distributed as follows:4790 in the SU ≥ 410 μmol/L and FLI < 80 group,4244 in the SU ≥ 410 μmol/L and FLI ≥ 80 group,24,402 in the SU < 410 μmol/L and FLI < 80 group, and4877 in the SU < 410 μmol/L and FLI ≥ 80 group.

The baseline characteristics of the study subjects by SU levels (≥410 and <410 mmol/L) and FLI values (<80 and ≥80) are shown in Table 1.

Figure 2 illustrates the correlation between FLI and SU in the study population, separately for women and men. The mean (SD) SU level was 330 μmol/L (78) in women and 387 μmol/L (86) in men, with the difference being statistically significant (*p* < 0.001). A moderate positive correlation was observed between SU concentration and FLI score in both women (r = 0.47) and men (r = 0.37).

Individuals with metabolic hyperuricemia had a significantly higher FLI than those with renal hyperuricemia, with mean (SD) values of 73.4 (12.2) and 69.6 (22.5), respectively (*p* = 0.015), after adjusting for sex and diabetes.

Table 2 shows the mortality risk stratified by renal function in hyperuricemic individuals with FLI < 80 and FLI ≥ 80.

Table 3 presents the adjusted hazard ratios for all-cause and cardiovascular mortality over the 15-year follow-up period, comparing individuals with FLI < 80 to those with FLI ≥ 80, and SU < 410 μmol/L to those with SU ≥ 410 μmol/L. Adjustments were made for age, sex, education, smoking status, alcohol consumption, body mass index, hypertension, dyslipidemia, and diabetes. The hazard ratios increased with higher FLI and SU levels. Notably, the combination of high FLI and high SU was associated with a 76% increase in all-cause mortality and a 70% increase in cardiovascular mortality compared to individuals with FLI < 80 and SU < 410 μmol/L.

Table 4 presents the number of subjects and cause-specific mortality ratios over the 15-year follow-up period, categorized by SU levels and FLI scores. Neoplasm-related mortality (ICD-10 codes C00-D89) was significantly lower in individuals with FLI < 80 compared to those with FLI ≥ 80. Cardiovascular mortality (ICD-10 codes I00-I99) was significantly lower in individuals with FLI < 80 and SU < 410 μmol/L compared to those with higher FLI and SU. Conversely, respiratory mortality was significantly higher in individuals with hyperuricemia, even if FLI was high.

No statistically significant differences were observed between the groups for mortality related to nervous system disorders (ICD-10 codes G00-G99), mental, behavioral, and neurodevelopmental disorders (ICD-10 codes F00-F99), gastrointestinal disorders (ICD-10 codes K00-K93), endocrine disorders (ICD-10 codes E00-E99), and external causes (ICD-10 codes V01-Y98). Additionally, alcohol-related mortality (ICD-10 categories F10, G31.2, G40.51, G62.1, G72.1, I42.6, K29.2, K70, K85.2, K86.0, O35.4, P04.3, Q86.0) did not differ significantly between the groups.

We also studied the standardized mortality ratio (SMR) related to SU concentrations and FLI scores (Figure 3). Mortality rates in GOAL participants with FLI score < 80 were lower than in the general population. Mortality rates in GOAL participants with FLI score ≥ 80 did not differ from the general population. In GOAL participants with SU ≥ 410 μmol/L, SMRs were higher than in individuals with lower SU levels.

## 4. Discussion

Our study corroborates previous reports that both hyperuricemia and hepatic steatosis are rather prevalent in the elderly population [38,39,40,41,42,43]. A positive correlation between hyperuricemia and hepatic steatosis has been reported in several studies, although most have been conducted in non-Western populations [24,25,26]. While the earliest known report of this association appears to be from Italy, studies from Western countries remain limited [44,45,46,47]. To the best of our knowledge, our study is the first to demonstrate this association in a Nordic population.

We found a moderate positive correlation between the SU concentration and the FLI score in both women and men. Our results also show that a combination of hyperuricemia and steatotic liver is much more hazardous with regard to mortality than either one of these pathological conditions alone. In our study population the increased mortality is not explained by alcohol consumption since we adjusted the mortality results for alcohol consumption, and mortality related to alcohol consumption did not differ statistically significantly between SU and FLI groups. There may be a causal relationship between serum urate and steatotic liver disease, although findings from Mendelian randomization studies have been inconsistent [48,49]. It has been hypothesized that uric acid leads to hepatic steatosis by triggering endothelial dysfunction, insulin resistance, oxidative stress and systemic inflammation. Research indicates that when hepatocytes are exposed to uric acid, they become subject to a risk of mitochondrial dysfunction and heightened lipogenesis. This would, ultimately, result in the accumulation of fat in the liver [50]. Urate oxidase knockout mice were recently reported to spontaneously develop hyperuricemia and dysregulated lipid metabolism, accompanied by abnormal hepatic fat accumulation, suggesting that elevated uric acid may contribute to liver fat accumulation [51].

A novel finding in our study is a more pronounced association between SU and FLI in individuals with metabolic hyperuricemia (elevated SU in the absence of renal impairment) than in individuals with renal hyperuricemia (elevated SU due to an inability of the kidneys to excrete urate efficiently enough). A few recent publications have, however, reported that the [SU]/[serum creatinine] ratio is associated with steatotic liver disease and that high SU in combination with low creatinine (i.e., better kidney function) is a risk factor for hepatic steatosis [52,53]. We have previously reported that metabolic hyperuricemia is associated with remarkably higher all-cause mortality and cardiovascular mortality [18,19] and have proposed that this might be due to the more pronounced toxicity of metabolic hyperuricemia—when uric acid is produced in excess from xanthine and hypoxanthine, reactive oxygen species (ROS) are produced in excess. ROS cause endothelial dysfunction leading to cardiovascular morbidity and ultimately to increased mortality. The same mechanism might explain how metabolic hyperuricemia increases the risk of steatotic liver, since oxidative stress and endothelial dysfunction are involved in the pathogenesis of steatotic liver [54,55]. Uric acid induces endothelial dysfunction also by activating the HMGB1/RAGE signaling pathway [56]. In renal hyperuricemia, uric acid is not produced in excess (it accumulates due to renal underexcretion) and ROS is not produced in excess. There is growing evidence that the harmful hepatic effects associated with hyperuricemia may arise not only from elevated serum urate levels but also from reactive byproducts generated during urate synthesis. In particular, ROS, produced via XOR activity, can contribute to oxidative stress and liver injury. Supporting this, Yagi et al. reported that plasma XOR activity is associated with hepatic steatosis independently of insulin resistance and serum urate levels [57]. Other studies have further linked XOR activity to liver disease progression [58,59]. These findings suggest that targeting XOR activity may be important for reducing liver-related morbidity and mortality.

Our findings contribute to the growing understanding of the pathogenic heterogeneity of metabolic dysfunction-associated steatotic liver disease (MASLD) and its relevance to precision medicine. A stronger association with increased FLI and all-cause mortality in metabolic versus renal hyperuricemia suggests that the source of hyperuricemia carries distinct clinical implications and supports the notion that MASLD is not a uniform disease entity but rather a condition with diverse metabolic drivers. Furthermore, the observed sex-specific correlations between SU and FLI reinforce the importance of sex as a biological modifier in MASLD pathogenesis. These results highlight the potential utility of SU and renal function profiling for individualized risk stratification in MASLD and underscore the need for further studies exploring targeted interventions based on metabolic versus renal contributors to hyperuricemia. Incorporating these distinctions into clinical practice could advance precision medicine approaches aimed at improving the management and outcomes of patients with MASLD.

Currently, we do not know if lowering SU pharmacologically would reduce the incidence of steatotic liver disease in humans, but SU has been speculated as a viable treatment target almost a decade ago [60]. The xanthine oxidase inhibitor febuxostat reduces the development of non-alcoholic steatohepatitis in rodent models [61,62]. Also, allopurinol and benzbromarone have been shown to be effective in ameliorating hepatic steatosis in an animal model [63]. Clinical trials are lacking. A recent study however found allopurinol to be effective in reduction of controlled attenuation parameter (CAP) score—an indicator of hepatic steatosis based on ultrasonic properties of retropropagated radiofrequency signals acquired by transient elastography [64]—in patients with hyperuricemia [65]. The same trend was observed in hyperuricemic patients treated with febuxostat; however, it did not reach statistical significance. According to gout treatment guidelines we should prescribe urate lowering therapy (ULT) to our gout patients and treat SU levels to target, especially in those patients who have had recurrent gout attacks. These patients might benefit from ULT not only due to the prevention of gout flares but also due to a reduction in the risk of significant steatotic liver disease. Current national treatment guidelines do not generally recommend treatment of asymptomatic hyperuricemia, with some exceptions (Polish Society of Hypertension guidelines [66], Japanese guideline for management of hyperuricemia and gout [67]). Usually, it is thought that the evidence for a favorable benefit/harm ratio is insufficient. Our study implies that ULT trials may be indicated for the population at risk of hepatic steatosis. Also, it would be plausible and beneficial to run such trials separately and specifically for individuals with metabolic hyperuricemia since this population would apparently benefit more from ULT than the population with renal hyperuricemia.

Our study is characterized by some notable strengths. The study was conducted in a population-based setting that closely mirrors the demographic composition of a certain age group in a Nordic country. Enrollment was relatively high, 66% of invited individuals participated, ensuring a robust representation of both women and men. Additionally, we meticulously adjusted the results of mortality for numerous potential confounding factors, enhancing the credibility and accuracy of our findings. Let it be noted that adjusting for BMI and waist circumference was unnecessary, as they are already factors in the FLI equation. The follow-up duration was no less than 15 years, which allowed for the identification of associations between the studied variables and mortality that might have remained hidden, had follow-up been shorter. Furthermore, for identification of individuals with steatotic liver in the study population we used a very accurate predictor of hepatic steatosis—the fatty liver index (FLI) published by Bedogni et al. To guarantee a highly reliable assessment of kidney function, both creatinine and cystatin C were used to estimate glomerular filtration.

Nevertheless, although the FLI is a reliable predictor of hepatic steatosis, it has limitations. Studies evaluating its ability to predict liver-related outcomes such as cirrhosis and hepatocellular carcinoma are limited, and its sensitivity and specificity are not perfect [68], which may lead to some over- or underdiagnoses of steatotic liver in our study. Unfortunately, we did not have the chance to validate the findings of steatotic liver with imaging or biopsy. It has also been suggested that the FLI could be improved by incorporating key modifiers of MASLD epidemiology [68] —one of which is sex. This is particularly relevant, as there is strong evidence that steatotic liver disease exhibits sexual dimorphism [69,70,71]. To highlight this difference, we present the correlation between serum urate SU levels and FLI values separately for women and men (Figure 2).

Although we adjusted for potential confounders, residual confounding may still be present, as in all observational studies. Regarding glucose metabolism, including C-peptide and hemoglobin A1C in the analysis would have been beneficial, but these data were unavailable.

One limitation of the study is that it is likely that the most severely ill and disabled individuals are underrepresented. This could be due to a higher likelihood of reluctance or inability of the most severely ill patients to participate. On the other hand, including more of the most seriously ill individuals in the study would probably only strengthen the observed associations between mortality on the one hand and hyperuricemia and steatotic liver on the other, since kidney dysfunction, elevated SU levels and hepatic steatosis might be even more prevalent in this group of patients. Survivorship bias is also likely to explain the relatively low SMRs of GOAL participants with and without hyperuricemia and steatotic liver disease.

We posit that the majority of individuals with hyperuricemia in our study population were asymptomatic. This assumption is based on the rare use of ULT among the study population during the first part of the follow-up period, which spanned from 2002 to 2012. It is, however, important to note that we have only limited information on the diagnoses of gout in these individuals.

A limitation of our study is the lack of information regarding diuretic use. Diuretics, especially thiazide diuretics, are known to increase SU levels [72]. In our study, more than 40% of hyperuricemic individuals were on some form of antihypertensive medication. Therefore, the proportion of participants using diuretics could be substantial. Although Table 1 provides data on antihypertensive medication use within our study population, it does not specify the proportion of individuals using diuretics compared to other antihypertensive drugs.

Another limitation pertains to our attempt to categorize hyperuricemia into two distinct types—metabolic and renal. This classification is based on the point prevalence of reduced or normal eGFR and elevated or normal SU levels as observed in a cross-sectional snapshot in the baseline year of 2002. Given that elevated SU has been associated with kidney function deterioration, it is conceivable that the same individual initially categorized as having metabolic hyperuricemia at baseline might later, due to worsening kidney function, be reclassified as having renal hyperuricemia, possibly for a significant portion of the follow-up period. Additionally, there may exist a subset of individuals with hyperuricemia of mixed etiology, i.e., who exhibit an overlap between both types of hyperuricemia.

## 5. Conclusions

Hyperuricemia correlates positively with hepatic steatosis, and urate may be involved in the pathogenesis of steatotic liver disease. Metabolic hyperuricemia—defined as elevated serum urate levels in the presence of normal renal function—appears to be more strongly associated with steatotic liver disease than renal hyperuricemia. While the observational nature of the study precludes conclusions about causality, our study provides supporting evidence for distinguishing between metabolic and renal hyperuricemia and underscores the need for further research aiming to reproduce the results of our study in different populations and evaluating other effects of uric acid separately in individuals with normal versus impaired kidney function. Elevated SU could serve as a promising treatment target for individuals at risk of developing hepatic steatosis and for those with incipient hepatic steatosis. This approach may be particularly suitable for individuals with near-to-normal renal function. Trials to assess the effectiveness of ULT in these individuals are clearly warranted. Such trials would clarify the balance between the benefits and risks of lowering urate levels as a preventive measure against significant hepatic steatosis.

## Figures and Tables

**Figure 1 metabolites-15-00356-f001:**
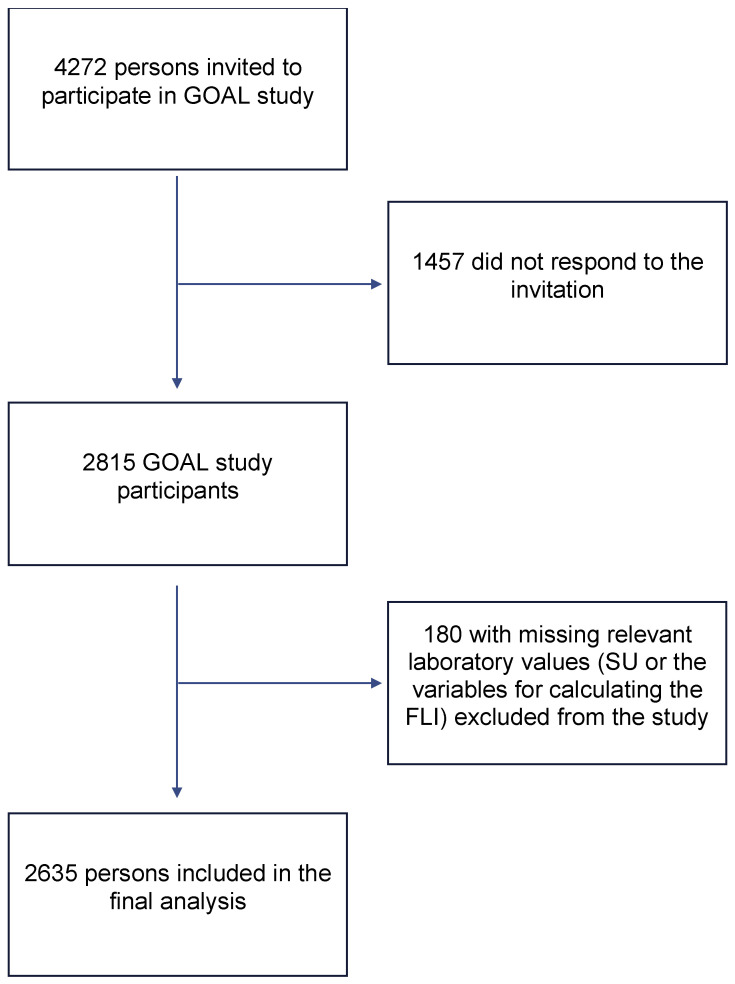
Flowchart detailing the selection of study subjects. *SU*, *serum urate*; *FLI*, *fatty liver index*.

**Figure 2 metabolites-15-00356-f002:**
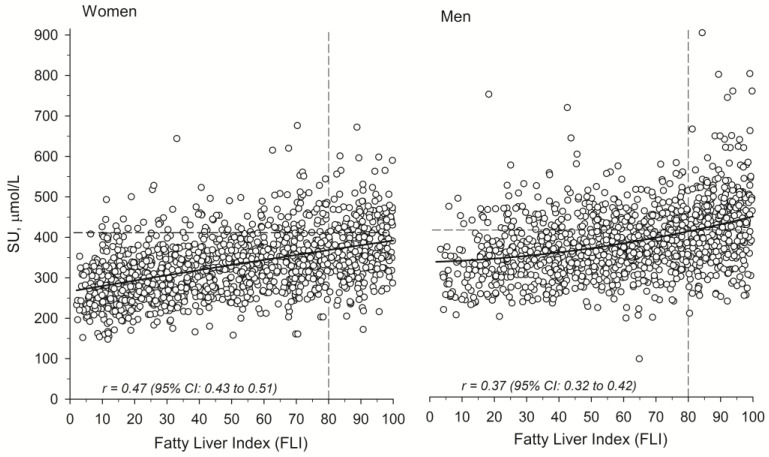
Correlation between SU levels and FLI value in women and men. The dashed lines represent the 75th percentile of SU and FLI. *SU*, *serum urate*; *FLI*, *fatty liver index*.

**Figure 3 metabolites-15-00356-f003:**
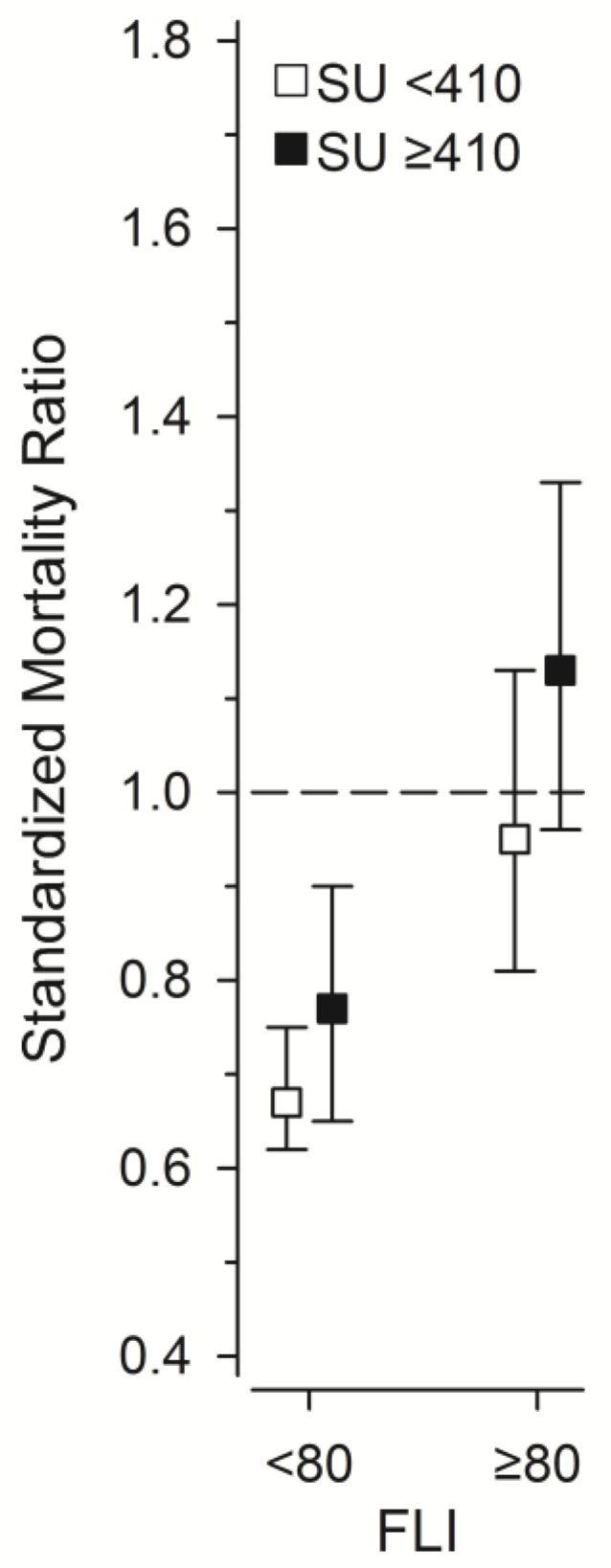
Standardized mortality ratio (SMR) related to SU concentrations and FLI scores. *SU*, *serum urate*; *FLI*, *fatty liver index.*

**Table 1 metabolites-15-00356-t001:** Baseline characteristics of the study subjects by SU (<410 μmol/L vs. ≥410 μmol/L) levels and FLI (<80 vs. ≥80) values.

	SU ≥ 410 μmol/L		SU < 410 μmol/L		*p*-Value		
	FLI < 80	FLI ≥ 80	FLI < 80	FLI ≥ 80	SU	FLI	Interaction
	*n = 346*	*n = 310*	*n = 1636*	*n = 343*			
Female, n (%)	110 (32)	101 (33)	996 (61)	179 (52)	<0.001	0.12	0.057
Age, mean (SD), years	64 (8)	62 (8)	61 (8)	62 (8)	0.007	0.071	<0.001
BMI, mean (SD), kg/m^2^	26.9 (2.7)	32.9 (4.7)	26.0 (3.2)	33.2 (4.4)	0.095	<0.001	<0.001
Waist circumference, mean (SD), cm							
Women	92 (7)	110 (10)	87 (9)	110 (9)	<0.001	<0.001	<0.001
Men	97 (8)	111 (9)	94 (8)	110 (9)	0.001	<0.001	0.14
WHtR	0.57 (0.05)	0.66 (0.06)	0.54 (0.06)	0.66 (0.06)	<0.001	<0.001	<0.001
LTPA, n (%)					0.018	<0.001	0.12
low	99 (29)	103 (34)	339 (21)	116 (35)			
moderate	143 (42)	131 (43)	696 (43)	128 (38)			
high	101 (29)	71 (23)	589 (36)	92 (27)			
BP, mmHg, mean (SD)							
systolic	147 (19)	152 (19)	144 (19)	151 (19)	0.045	<0.001	0.34
diastolic	86 (10)	90 (10)	85 (10)	90 (10)	0.23	<0.001	0.29
MAP	106 (11)	111 (11)	104 (11)	110 (11)	0.068	<0.001	0.25
Fasting plasma glucose, mean (SD), mmol/L	5.73 (1.06)	6.32 (1.48)	5.51 (1.13)	6.32 (1.81)	0.073	<0.001	0.081
Fasting plasma insulin, mean (SD)	10.0 (8.9)	17.7 (18.2)	8.1 (8.0)	14.2 (13.2)	<0.001	<0.001	0.60
Homa-IR, mean (SD)	2.70 (3.16)	5.19 (6.10)	2.08 (2.98)	4.29 (5.75)	<0.001	<0.001	0.67
Cholesterol, mean (SD), mmol/L	5.62 (1.17)	5.83 (1.14)	5.76 (1.05)	5.80 (1.16)	0.27	0.015	0.11
HDL, mean (SD), mmol/L	1.43 (0.40)	1.28 (0.35)	1.60 (0.44)	1.35 (0.39)	<0.001	<0.001	0.018
LDL, mean (SD), mmol/L	3.50 (1.05)	3.56 (0.99)	3.57 (0.93)	3.54 (0.99)	0.57	0.73	0.33
Triglycerides, mean (SD), mmol/L	1.50 (0.71)	2.33 (2.08)	1.30 (0.60)	2.01 (0.95)	<0.001	<0.001	0.23
eGFR, mean (SD), ml/min/1.73 m^2^	69.2 (16.0)	72.1 (17.1)	80.0 (14.2)	79.3 (15.0)	<0.001	0.12	0.014
Cystatin C, mean (SD), mg/L	1.12 (0.25)	1.10 (0.25)	0.96 (0.17)	1.01 (0.18)	<0.001	0.091	<0.001
GGT, mean (SD), U/L	43.1 (35.5)	80.5 (87.4)	34.1 (31.5)	73.6 (118.8)	0.007	<0.001	0.72
ASAT, mean (SD), U/L	32 (14)	38 (18)	29 (9)	34 (22)	<0.001	<0.001	0.70
ALAT, mean (SD), U/L	12 (9)	16 (13)	10 (6)	15 (17)	0.002	<0.001	0.63
hsCRP, mean (SD), mg/L	3.1 (4.3)	4.1 (6.7)	2.3 (5.2)	3.6 (4.5)	0.013	<0.001	0.71
25(OH)D, mean (SD), nmol/L	36 (15)	31 (17)	36 (15)	32 (14)	0.71	<0.001	0.91
AUDIT score, mean (SD)	3.5 (2.6)	4.2 (2.9)	2.9 (2.3)	3.4 (2.8)	<0.001	<0.001	0.32
Smoking, n (%)	49 (14)	53 (17)	263 (16)	70 (20)	0.16	0.050	0.79
Education years, mean (SD)	9.2 (3.3)	9.4 (3.4)	9.6 (3.3)	9.0 (3.1)	0.98	0.18	0.013
Comorbidities, n (%)							
DM	36 (10)	77 (25)	120 (7)	79 (23)	0.075	<0.001	0.30
Hypertension	138 (40)	161 (52)	451 (28)	160 (47)	<0.001	<0.001	0.084
Musculoskeletal	123 (36)	119 (38)	551 (34)	138 (40)	0.98	0.046	0.43
CVD	49 (14)	42 (14)	128 (8)	38 (11)	0.003	0.27	0.15
Respiratory	26 (8)	36 (12)	106 (6)	29 (8)	0.14	0.027	0.58
MetS	185 (53)	283 (91)	528 (32)	300 (87)	<0.001	<0.001	0.098
Medication, n (%)							
Antihypertensive *	133 (38)	144 (46)	367 (22)	145 (42)	<0.001	<0.001	0.003
Lipid lowering *	77 (22)	58 (19)	224 (14)	67 (20)	0.031	0.40	0.009
Urate lowering **	35 (10)	59 (19)	27 (2)	12 (3)	<0.001	<0.001	0.94
Cohabiting, n (%)	255 (74)	220 (71)	1193 (73)	219 (64)	0.090	0.008	0.18

*SU, serum urate; FLI, fatty liver index; BMI, body mass index; WHtR, waist-to-height ratio; LTPA, Leisure-time physical activity, graded as low (exercise less than once a week), moderate (one to two times a week) or high (at least three times a week); BP, blood pressure; MAP, mean arterial pressure; HOMA-IR, homeostatic model assessment for insulin resistance; HDL, high-density lipoprotein; LDL, low-density lipoprotein; eGFR, estimated glomerular filtration rate, calculated using the CKD-EPI creatinine-cystatin C equation; GGT, gamma-glutamyl transferase; ALAT, alanine aminotransferase; ASAT, aspartate aminotransferase; hsCRP, high-sensitivity C-reactive protein; 25(OH)D, 25-hydroxyvitamin D; AUDIT, Alcohol Use Disorders Identification Test; DM, diabetes mellitus; CVD, cardiovascular disease; MetS, metabolic syndrome, defined by 2009 harmonized criteria.* * Number of patients who used medication at baseline. ** Number of patients with a history of a urate-lowering drug purchase in the active follow-up period of the study (years 2002–2012).

**Table 2 metabolites-15-00356-t002:** Adjusted hazard ratio (HR) for all-cause mortality stratified by renal function in 15-year follow-up period in hyperuricemic individuals with FLI < 80 and FLI ≥ 80.

	eGFR > 67HR (95% CI) *	eGFR ≤ 67HR (95% CI) *
FLI < 80	1.00 (Reference)	1.89 (1.31 to 2.73)
FLI ≥ 80	2.30 (1.31 to 3.14)	2.28 (1.49 to 3.50)

*eGFR*, *estimated glomerular filtration rate*, *calculated using the CKD-EPI creatinine-cystatin C equation*; *FLI*, *fatty liver index*. * Adjusted for age, sex, education, smoking status, alcohol consumption, body mass index, hypertension, dyslipidemia and diabetes.

**Table 3 metabolites-15-00356-t003:** Adjusted hazard ratio (HR) for all-cause and cardiovascular mortality in 15-year follow-up period.

	SU < 410 μmol/LHR (95% CI) *	SU ≥ 410 μmol/LHR (95% CI) *
All-cause mortality		
FLI < 80	1.00 (Reference) **	1.16 (0.95 to 1.40)
lFLI ≥ 80	1.34 (1.06 to 1.70)	1.76 (1.39 to 2.23)
Cardiovascular mortality		
FLI < 80	1.00 (Reference) **	1.16 (0.86 to 1.56)
FLI ≥ 80	1.26 (0.88 to 1.80)	1.70 (1.20 to 2.41)

*SU*, *serum urate*; *FLI*, *fatty liver index*. * Adjusted for age, sex, education, smoking status, alcohol consumption, body mass index, hypertension, dyslipidemia and diabetes. ** Denominator of hazard ratios.

**Table 4 metabolites-15-00356-t004:** Numbers [n] and age- and sex-adjusted mortality per 1000 person–years (95% CI) of cause-specific deaths during the 15-year follow-up period according to SU and FLI levels.

	SU ≥ 410 μmol/L		SU < 410 μmol/L		
	FLI < 80 (Group 1)	FLI ≥ 80 2 (Group 2)	FLI < 80 3 (Group 3)	FLI ≥ 80 4 (Group 4)	*p*-Value [*]
Cancer and other neoplasms (ICD codes C00-D89)	[31] 5.5 (3.6 to 7.5)	[35] 8.4 (5.6 to 11.1)	[128] 6.1 (5.0 to 7.1)	[46] 10.1 (7.2 to 13.0)	0.009 [1/4, 3/4]
Cardiovascular (ICD codes I00-I99) *	[61] 10.4 (7.8 to 13.1)	[70] 17.6 (13.5 to 21.8)	[176] 8.8 (7.5 to 10.1)	[63] 14.7 (11.1 to 18.4)	<0.001 [1/2, 2/3, 3/4]
Nervous system (ICD codes G00-G99) *	[11] 2.0 (0.8 to 3.2)	[13] 3.6 (1.6 to 5.5)	[74] 3.6 (2.8 to 4.4)	[10] 2.3 (0.9 to 3.7)	0.20
Respiratory system (ICD codes J00-J99) *	[12] 2.0 (0.9 to 3.2)	[7] 1.6 (0.4 to 2.8)	[15] 0.7 (0.4 to 1.1)	[2] 0.4 (0.0 to 1.1)	0.030 [1/4]
Mental, behavioral and neurodevelopmental disorders (ICD codes F00-F99) *	[3] 0.5 (0.0 to 1.1)	[5] 1.4 (0.1 to 2.6)	[9] 0.5 (0.2 to 0.8)	[3] 0.7 (0.0 to1.6)	0.26
Gastrointestinal (ICD codes K00-K93) *	[6] 1.1 (0.2 to 2.0)	[4] 0.9 (0.0 to 1.8)	[13] 0.6 (0.3 to 0.9)	[3] 0.6 (0.0 to 1.3)	0.59
Endocrine (ICD codes E00-E99) *	[3] 0.5 (0.0 to 1.1)	[4] 0.8 (0.0 to 1.7)	[3] 0.1 (0.0 to 0.3)	[3] 0.6 (0.0 to 1.4)	0.13
External causes (ICD codes V01-Y98)	[3] 0.5 (0.1 to 1.1)	[1] 0.2 0.0 to 0.6)	[15] 0.7 (0.3 to 1.1)	[3] 0.7 (0.0 to 1.4)	0.70
Alcohol-related **	[1] 0.4 (0.0 to 1.1)	[3] 0.6 (0.0 to 1.3)	[2] 0.1 (0.0 to 0.2)	[2] 0.4 (0.1 to 0.9)	0.20

*SU*, *serum urate*, *μmol/L*; *FLI*, *fatty liver index.* * Sidak’s multiple comparison procedure was used to correct significance levels for post hoc testing (p50.05). ** Alcohol-related causes of death included all diseases caused by alcohol (ICD-10 categories F10, G31.2, G40.51, G62.1, G72.1, I42.6, K29.2, K70, K85.2, K86.0, O35.4, P04.3, Q86.0).

## Data Availability

The data presented in this study are available on request from the corresponding author due to legal and ethical reasons.

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
