# Peer review of "Hyperuricemia—Especially “Metabolic Hyperuricemia”—Is Independently Associated with a Higher Risk of Steatotic Liver Disease"

_metabolites, 2025, doi:10.3390/metabo15060356_

Round 1
Reviewer 1 Report
Comments and Suggestions for Authors
GENERAL COMMENT
I enjoyed reading this innovative piece of research which adds another (renal) piece in the complex puzzle relating steatotic liver disease (SLD) with raised serum uric acid levels. This line of research is consistent with the emerging notion of Cardiovascular-Kidney-Metabolic Health (PMID: 37807924) in the context of which SLD is positioned as an upstream risk factors eventually culminating in cardiovascular morbidity and mortality in a subset of individuals. However, the study design needs clarification and disease nomenclatures should be updated.
SPECIFIC COMMENT
MAJOR
- Introduction is not particularly focused and must be reworked.
- To this end, these authors may be willing to explain the association linking SLD with raised SUO levels first. Please note that, while an exhaustive review of literature is not necessary here, as a matter of fact a robust line of research documents this association exhaustively (PMID: 28790228; PMID: 28244714; PMID: 26738417; PMID: 26395162; PMID: 26308292; PMID: 28244714). The contention that these studies were based In Asia is not fully convincing unless these Authors explain why Finnish should be biologically different from Asians regarding the association between SLD and raised SUA levels. Incidentally it is noted that the first study showing an independent association between what we now call MASLD and SUA was published in Italy in 2002.
- The manuscript should explain the element of novelty in the present study, i.e. introduce the notion of “metabolic hyperuricemia”.
- Statement of the research question: what did these Authors expect to find and why?
- Given that most of these patients are generally asymptomatic, the allusions to gout (and gout’s history) may be deleted here. However, these notions may possibly be utilized elsewhere in the manuscript based on Authors’ preferences.
- Method
- By checking in PubMed database on the 18th of April “metabolic hyperuricemia [Title/Abstract]” only one published was retrieved [PMID: 39768539] which is authored by the same group of Researchers indicating that this novel notion of metabolic hyperuricemia is not completely arbitrary but is not (yet) universally accepted. Therefore, these Authors may be willing to explain better the rationale underlying the reason why, in the present study they define hyperuricemia as renal if estimated glomerular filtration 28 rate (eGFR) was ≤67 ml/min/1,73 m2 (25th percentile) and as metabolic if eGFR was >67 29 ml/min/1,73 m2.
- Through the manuscript these authors use the outdated definition of “Fatty liver”. Please, be aware that a Consensus Conference proposed novel disease definitions in this field (PMID: 37363821) and these are presently almost universally accepted. It is true that investigators may retain the freedom to use the nomenclatures they deem to fit their studies best. However, it would be advisable to explain why they want to use “fatty liver”.
- Discussion
- A valuable adjunct at lines 294-295 would be discussing PMID: 35363278.
- At lines 357-360 please note that Professor Bedogni et al. have illustrated the limitations of the Fatty Liver Index (DOI: 20517/mtod.2021.08).
- SLD is a sexually dimorphic disease (PMID: 38890029). This should be further emphasized given that the authors conducted sex-specific analysis of data, in agreement with current views (PMID: 33982400, PMID: 30924946).
- Considering the methodological limitations of the study it would be worth explaining that the present study may be a proof-of-evidence investigation stimulating additional research in the field of “metabolic hyperuricemia”.
- Discuss the potential impact of this study in the context of MASLD pathogenic heterogeneity and its inherent impact in precision medicine approaches.
MINOR
- Based on my comment 1.a) above, lines 281-284 from “The association” to “hepatic steatosis” should be deleted.
- Throughout the manuscript “serum uric acid” is sometimes abbreviated “SU” and other times the word “SUA” is used (e.g. Table 1). It would be better to avoid any discrepancies.
- Line 95. Use the same font as for other headings.
Author Response
Comment: Introduction is not particularly focused and must be reworked.
Response: We have revised the introduction in accordance with your suggestions in the updated manuscript.
Comment: To this end, these authors may be willing to explain the association linking SLD with raised SUO levels first. Please note that, while an exhaustive review of literature is not necessary here, as a matter of fact a robust line of research documents this association exhaustively (PMID: 28790228; PMID: 28244714; PMID: 26738417; PMID: 26395162; PMID: 26308292; PMID: 28244714). The contention that these studies were based In Asia is not fully convincing unless these Authors explain why Finnish should be biologically different from Asians regarding the association between SLD and raised SUA levels. Incidentally it is noted that the first study showing an independent association between what we now call MASLD and SUA was published in Italy in 2002.
Response: Thank you for your comment. The association between serum urate levels and steatotic liver disease has indeed been demonstrated in previous studies, many of which have been summarized in recent meta-analyses. To avoid redundancy, we refer to the most recent and comprehensive meta-analysis in our manuscript (reference 26). Notably, most prior studies on this association were conducted in Asian populations. To the best of our knowledge, no similar studies have been carried out in Nordic populations, making our findings novel and particularly relevant given the differences in the prevalence of steatotic liver disease and hyperuricemia, as well as genetic differences in urate metabolism between Asian and Western populations. We are grateful for your reference to the first study demonstrating this association; we have now cited it in the revised manuscript (reference 44).
Comment: The manuscript should explain the element of novelty in the present study, i.e. introduce the notion of “metabolic hyperuricemia”.
Response: The concepts of metabolic and renal hyperuricemia are introduced in the Introduction, and their association with higher fatty liver index FLI in individuals with metabolic hyperuricemia compared to those with renal hyperuricemia is demonstrated in the Results section and further discussed in the Discussion.
Comment: Statement of the research question: what did these Authors expect to find and why?
Response: The research question is added to the Introduction in the revised manuscript (page 2, lines 88-94).
Comment: Given that most of these patients are generally asymptomatic, the allusions to gout (and gout’s history) may be deleted here. However, these notions may possibly be utilized elsewhere in the manuscript based on Authors’ preferences.
Response: In the revised manuscript, allusions to gout have been minimized and historical context removed, resulting in a more concise and focused introduction.
Comment: By checking in PubMed database on the 18th of April “metabolic hyperuricemia [Title/Abstract]” only one published was retrieved [PMID: 39768539] which is authored by the same group of Researchers indicating that this novel notion of metabolic hyperuricemia is not completely arbitrary but is not (yet) universally accepted. Therefore, these Authors may be willing to explain better the rationale underlying the reason why, in the present study they define hyperuricemia as renal if estimated glomerular filtration 28 rate (eGFR) was ≤67 ml/min/1,73 m2 (25th percentile) and as metabolic if eGFR was >67 29 ml/min/1,73 m2.
Response: We defined hyperuricemia as renal when the estimated glomerular filtration rate (eGFR) was ≤67 ml/min/1.73 m² (25th percentile), and as metabolic when eGFR was >67 ml/min/1.73 m². The threshold is based on percentiles to reflect the characteristics of our specific study population. This classification is consistent with previous studies that have used a similar distinction. The rationale for this approach is detailed in the Methods section (page 4, lines 146–150).
Comment: Through the manuscript these authors use the outdated definition of “Fatty liver”. Please, be aware that a Consensus Conference proposed novel disease definitions in this field (PMID: 37363821) and these are presently almost universally accepted. It is true that investigators may retain the freedom to use the nomenclatures they deem to fit their studies best. However, it would be advisable to explain why they want to use “fatty liver”.
Response: Thank you for your comment. In the revised manuscript, we have adopted comprehensive and current liver disease nomenclature.
Comment: A valuable adjunct at lines 294-295 would be discussing PMID: 35363278.
Response: Thank you for your valuable suggestion. In the revised manuscript, we have incorporated a discussion of the results from Mendelian randomization studies examining the association between serum urate and steatotic liver disease (page 11, lines 330–332), and have added the reference you recommended (reference 49).
Comment: At lines 357-360 please note that Professor Bedogni et al. have illustrated the limitations of the Fatty Liver Index (DOI: 20517/mtod.2021.08).
Response: Thank you for the suggestion. In the revised manuscript, we have cited the recommended paper (reference 68) when discussing the limitations of the fatty liver index (page 13, lines 415–421).
Comment: SLD is a sexually dimorphic disease (PMID: 38890029). This should be further emphasized given that the authors conducted sex-specific analysis of data, in agreement with current views (PMID: 33982400, PMID: 30924946).
Response: We have emphasized the sexually dimorphic nature of steatotic liver disease in the revised manuscript (page 13, lines 420–424) and have added relevant references to support this point (references 69-71).
Comment: Considering the methodological limitations of the study it would be worth explaining that the present study may be a proof-of-evidence investigation stimulating additional research in the field of “metabolic hyperuricemia”.
Response: Thank you for your comment. Following your suggestion, we have framed our study as one that provides supporting evidence for the need for further research distinguishing between metabolic and renal hyperuricemia (page 14, lines 464-469).
Comment: Discuss the potential impact of this study in the context of MASLD pathogenic heterogeneity and its inherent impact in precision medicine approaches.
Response: Thank you for your comment. In the revised manuscript, we have expanded the discussion to address the potential implications of our findings in the context of MASLD pathogenic heterogeneity and their relevance for precision medicine approaches (page 12, lines 365–377).
Comment: Based on my comment 1.a) above, lines 281-284 from “The association” to “hepatic steatosis” should be deleted.
Response: In response to your previous comment, we have removed the mentioned lines of the original paragraph and rephrased the discussion of prior studies on the association between hyperuricemia and hepatic steatosis.
Comment: Throughout the manuscript “serum uric acid” is sometimes abbreviated “SU” and other times the word “SUA” is used (e.g. Table 1). It would be better to avoid any discrepancies.
Response: Thank you for pointing this out. In the revised manuscript, we have addressed this inconsistency by using the term serum urate (SU) consistently throughout the text and tables to ensure clarity and uniformity.
Comment: Line 95. Use the same font as for other headings.
Response: Thank you for pointing out the formatting mistake. We have corrected it in the revised manuscript.
Reviewer 2 Report
Comments and Suggestions for Authors
This study addresses a clinically relevant and emerging topic, such as the association between hyperuricemia and hepatic steatosis. The authors appropriately highlight the potential pathogenic role of elevated uricemia, especially in metabolic hyperuricemia. The number of the studied sample is meaningful. This distinction between metabolic and renal hyperuricemia is insightful and contributes to the understanding of risk stratification in NAFLD. However, a clarification should be made that the current evidence shows mainly correlation rather than causation. Thus, the role of uric acid as an independent driver of steatosis remains a credible hypothesis
In line 95, the Materials and Methods should be written in the same font as any section title.
The flowchart detailing the selection of study subjects on line 190 should be included in the Materials and Methods because it shows the conceptualization of the study.
Author Response
Comment: This study addresses a clinically relevant and emerging topic, such as the association between hyperuricemia and hepatic steatosis. The authors appropriately highlight the potential pathogenic role of elevated uricemia, especially in metabolic hyperuricemia. The number of the studied sample is meaningful. This distinction between metabolic and renal hyperuricemia is insightful and contributes to the understanding of risk stratification in NAFLD. However, a clarification should be made that the current evidence shows mainly correlation rather than causation. Thus, the role of uric acid as an independent driver of steatosis remains a credible hypothesis
Response: Thank you for your comment. In the revised manuscript, we have adopted more cautious language in our conclusions, emphasizing that the observational nature of the study identifies associations rather than causality, and thus precludes definitive conclusions about cause-and-effect relationships.
Comment: In line 95, the Materials and Methods should be written in the same font as any section title.
Response: Thank you for pointing out the formatting mistake. We have corrected it in the revised manuscript.
The flowchart detailing the selection of study subjects on line 190 should be included in the Materials and Methods because it shows the conceptualization of the study.
Response: In the revised manuscript the flowchart detailing the selection of study subjects is included in the Materials and Methods.
Reviewer 3 Report
Comments and Suggestions for Authors
Reviewers' comments:
I reviewed the manuscript, metabolites-3613973 ‘Hyperuricemia -Especially "Metabolic Hyperuricemia" -Is Independently Associated with Higher Risk of Fatty Liver’. Author described that metabolic hyperuricemia (normal renal function) is more hazardous for all-cause and mortality. I think that this manuscript was very interested. However, I feel that a little more detailed explanation is needed. Please consider making some corrections and additions.
- In generally, there are AST, ALT and gamma-GTP in the biomarker of liver function. In Table 1, how was AST and ALT? Author should add their data in Table 1. Moreover, how was CRP and 25-hydroxyvitamin D?
- Medication in Table1, how was urate lowering drugs? Did you exclude? Author should add about anti-hyperuricemia agents.
- The concentration of blood uric acid in male is generally higher than that of female. How was statistically sex-difference in Figure 2? Author should describe in gender difference.
- Line 128, author described that the cut-off value of FLI was used 30 and 60. Why did author apply 80 not 60 in LFI value?
- Line 58, author use a term of 'XO'. This has been frequently used from discovery by Schardinger in 1902 or Morgan in 1920. However, in recently, a number of article (original paper or review) has been utilized XOR not XO due to be understood the molecular mechanism of xanthine oxidoreductase. Thus, I recommend that author should correct to xanthine oxidoreductase(XOR), and add the feature of XOR.
- In tableï¼’ to 4, author shows that FLI>80 in regardless of eGFR is hazardous for all-cause mortality, especially cardiovascular mortality. Dose these results mean that fatty liver would be easy to lead to cardiovascular disease not hepatic fibrogenesis, hepatic carcinoma, and cirrhosis?
- Author describes that metabolic hyperuricemia is correlate with all-cause mortality due to the pronounced toxicity by XO induced ROS when uric acid was produced. Yagi and Kusunoki et al., reported that plasma XOR activity is associated with hepatic steatosis independent of insulin resistance and serum UA levels. Sato K., et.al., and Kawachi Y., et.al., reported that plasma XOR activity is associated with the progression of liver disease. To suppress from mortality or progression of hepatic steatosis, which is urate lowering and inhibition of XOR activity serious? Please describe in discussion part.
Author Response
Comment: In generally, there are AST, ALT and gamma-GTP in the biomarker of liver function. In Table 1, how was AST and ALT? Author should add their data in Table 1. Moreover, how was CRP and 25-hydroxyvitamin D?
Response: In the revised manuscript, we have added information on aspartate aminotransferase, alanine aminotransferase, gamma-glutamyl transferase, high-sensitivity C-reactive protein, and 25-hydroxyvitamin D to Table 1.
Comment: Medication in Table1, how was urate lowering drugs? Did you exclude? Author should add about anti-hyperuricemia agents.
Response: A small proportion of our study population used urate-lowering drugs. This information has been added to Table 1 in the revised manuscript.
Comment: The concentration of blood uric acid in male is generally higher than that of female. How was statistically sex-difference in Figure 2? Author should describe in gender difference.
Response: We acknowledge the sex-related difference in serum urate (SU) concentrations. In the revised manuscript, we have included information on this between-sex difference in SU levels in our study population (page 7, lines 227–228).
Comment: Line 128, author described that the cut-off value of FLI was used 30 and 60. Why did author apply 80 not 60 in LFI value?
Response: We acknowledge that the originally proposed cut-off values for the fatty liver index (FLI) are 30 and 60. However, the same study also demonstrated that using a cut-off of 80 increases the specificity of FLI to 96%. Given the high prevalence of hepatic steatosis in our study population, and the fact that an FLI value of 80 corresponded to the 75th percentile within our cohort, we found it reasonable to adopt this higher threshold in our analyses.
Comment: Line 58, author use a term of 'XO'. This has been frequently used from discovery by Schardinger in 1902 or Morgan in 1920. However, in recently, a number of article (original paper or review) has been utilized XOR not XO due to be understood the molecular mechanism of xanthine oxidoreductase. Thus, I recommend that author should correct to xanthine oxidoreductase(XOR), and add the feature of XOR.
Response: Thank you for your suggestion. You are correct that “xanthine oxidoreductase” is a more accurate term than “xanthine oxidase” in the context of our work. We have revised the manuscript accordingly to consistently use the term “xanthine oxidoreductase.”
Comment: In tableï¼’ to 4, author shows that FLI>80 in regardless of eGFR is hazardous for all-cause mortality, especially cardiovascular mortality. Dose these results mean that fatty liver would be easy to lead to cardiovascular disease not hepatic fibrogenesis, hepatic carcinoma, and cirrhosis?
Response: Thank you for your question. In Tables 2 and 3, we present the effects of interactions between fatty liver index (FLI) and estimated glomerular filtration rate (eGFR), and between FLI and serum urate (SU), respectively, on mortality. However, the results of our study do not allow us to infer that fatty liver more readily leads to cardiovascular disease than to liver-related outcomes such as hepatic fibrogenesis, hepatocellular carcinoma, or cirrhosis. The number of gastrointestinal deaths in our study population is reported in Table 4; there were 26 gastrointestinal deaths in total, of which only 8 were due to liver diseases. Importantly, the associations observed between FLI and cause-specific mortality in our study are correlational and do not establish causality.
Comment: Author describes that metabolic hyperuricemia is correlate with all-cause mortality due to the pronounced toxicity by XO induced ROS when uric acid was produced. Yagi and Kusunoki et al., reported that plasma XOR activity is associated with hepatic steatosis independent of insulin resistance and serum UA levels. Sato K., et.al., and Kawachi Y., et.al., reported that plasma XOR activity is associated with the progression of liver disease. To suppress from mortality or progression of hepatic steatosis, which is urate lowering and inhibition of XOR activity serious? Please describe in discussion part.
Response: Thank you for your valuable comment. In the revised manuscript, we have added information regarding the significant role of xanthine oxidoreductase (XOR) activity in the development and progression of hepatic steatosis and liver disease (page 12, lines 357–364), along with the relevant references (references 57–59).
Round 2
Reviewer 1 Report
Comments and Suggestions for Authors
The collaborative attitude shown by these Colleagues is appreciated. As a result of this approach, the revised version of the manuscript is improved.